# The Spatio-Temporal Dynamic Patterns of Shallow Groundwater Level and Salinity: The Yellow River Delta, China

**Xiaomei Fan \*, Tong Min**  **and Xiaojie Dai**

School of Geographic Science, Nanjing University of Information Science & Technology, Pukou District, Nanjing 210044, China
* Correspondence: fanxm@lreis.ac.cn

**Abstract:** Shallow groundwater in coastal aquifers is a highly dynamic and complex system with a high risk of seawater intrusion. Analyzing the spatio-temporal dynamic patterns of groundwater can help to manage the groundwater resource and prevent it from degradation. Based on the groundwater level (GWL) and electrical conductivity (EC) monitoring data of 18 observation wells in the Yellow River Delta (YRD) from 2004 to 2010, this research analyses the groundwater dynamics using a robust seasonal trend decomposition technique (STL) and spatial interpolation method to detect the groundwater spatio-temporal dynamic patterns of groundwater level and salinity. Combined with hydro-climatic data, the Pearson correlation method and the Mann-Kendall (MK) trend analysis were used to further reveal the impacts that induce their trends and seasonal variations. Our analyses show that the risk of seawater intrusion into local shallow aquifers in this region is high, with the mean groundwater level over 42% of the region lower than the local sea level, and the mean groundwater EC over 96% of the region met the standards for seawater intrusion. In addition, the trends of groundwater level generally declined by 0.01~0.45 m/a and salinity increased by 1.153~25.608 μs/cm.a, which are consistent with the trend of precipitation decline. The seasonal dynamics of groundwater level and salinity are highly correlated with the seasonal components of rainfall and evaporation. It can be concluded that the extent of seawater intrusion will increase in the future with sea level rise. The approaches used in this study proved to be effective and can certainly serve as an example for the analysis of the spatio-temporal dynamics of groundwater in other coastal regions.

**Keywords:** groundwater level; electrical conductivity; spatio-temporal analysis; Yellow River Delta; seawater intrusion



## 1. Introduction

Shallow groundwater plays an important role in the coastal waters, not only because it can provide reliable fresh water for drinking and irrigation, but also because it has a close relationship with the terrestrial ecosystem [1,2]. However, the coastal groundwater system is very fragile under the threat of seawater intrusion and human activities. Once groundwater degradation occurs, it will affect the safety of drinking water in coastal areas and the stability of local agriculture and ecosystems [3]. It is important for nature managers to know the spatio-temporal dynamic patterns and quality of groundwater levels in the coastal regions.

Obtaining the spatial and temporal variation characteristics of groundwater is one of the essential problems of groundwater resource management. Recently, groundwater numerical simulation models, which have been applied to aquifers all over the world, are considered to be the best method that can give a complete insight into the groundwater system and predict its future development [4–6]. Using numerical modelling methods, Paniconi et al. [7] proposed that intensive groundwater pumping in this coastal aquifer leads to a regional decrease in piezometric head, resulting in seawater intrusion. However, the

best calibrated simulation model is still a highly idealized representation of the real system due to difficulties in accurate parameter estimation and simplifying model assumptions [8]. In addition, the construction of a groundwater numerical model is based on a complete understanding of the hydrogeological condition of the local aquifer. Actually, most of the rural areas in China lack the data of a hydrological survey, which limits the application of the groundwater model. As a result, some convenient and effective methods have been used to analyze the spatio-temporal variability of groundwater level (GWL) and electrical conductivity (EC), such as statistical methods and the spatial interpolation method. The most popular non-parametric method for analyzing the trend in the time series is the Mann-Kendall test [9–11]. The Mann-Kendall test can be used not only for hydrological series in general, but also for groundwater level series in particular [12]. A number of research studies using the Mann-Kendall test and Sen's slope estimator investigate the trends in annual, seasonal, and monthly groundwater levels as well as the future trend prediction of groundwater [13,14].

However, a major drawback of the traditional trend test procedure is that it can only identify the trends in the time series, but cannot characterize them [15]. Structural time series analysis provides an alternative that can simultaneously identify trends, seasonal components and autoregressive properties. Seasonal-trend decomposition can help to identify the similar and different shape and variability components between one well and another [16]. Structural time series analysis as implemented by Koopman and others was used to identify trend, seasonal and autoregressive components in quarterly water level measurements in Huron County between 1993 and 1997 [17]. Shamsudduha et al. [15] applied the seasonal-trend decomposition (STL) procedure to observations in the Ganges-Brahmaputra-Meghna Delta, detecting both seasonal patterns and long-term changes. This novel application of STL successfully reveals the uncertainty of groundwater-fed irrigation and the hydrological impact of potential seawater intrusion into coastal aquifers. The seasonal variation of coastal waters is usually analyzed in monthly or quarterly observation units [18,19]. Iqbal et al. divided the study area into wet and dry zones, and analyzed the relationship between water quality and NDVI at the annual and seasonal scales [20]. Eltahir et al. analyzed the abnormal fluctuation of aquifer water level from extreme drought and precipitation events, and found that precipitation and evaporation had a certain influence on the regional water cycle from the seasonal time scale [21]. However, seasonal changes in groundwater based on such analyses are often unsatisfactory [22]. The STL method used in this article can handle any type of seasonal data, not just monthly and quarterly data. Unlike traditional time series methods, it can be immune to outliers.

Water level and water quality are the most active factors in hydrodynamics and monitoring their dynamic changes is helpful to the management and utilization of coastal water resources [23]. Some studies on seawater intrusion have focused on groundwater levels [24,25]. One of the reasons for this is that obtaining time-continuous groundwater quality records is quite difficult due to time and cost constraints. However, in most coastal areas and deltas, understanding the dynamics and trends of groundwater quality (especially salinity) is important for an accurate assessment of seawater intrusion. Since deltas are prone to salinization due to anthropogenic changes in the hydrological cycle [26,27], any stimulation will lead to freshening or salinization on a seasonal timescale. Other studies focus on the freshwater–saltwater interface [28], which is far below the surface of the ground, while rare studies are concerned with shallow groundwater salinity and its spatial and temporal dynamics.

The specific objectives of this study included: (1) interpolating spatially and temporally varying groundwater levels and salinity on the YRD to assess the extent and severity of seawater intrusion. (2) To apply time-series decomposition (STL) and trend analysis techniques to quantify the evolving trends and seasonal dynamics of groundwater levels and salinity. (3) To analyze the main features (natural or anthropogenic) primarily responsible for the variable groundwater levels and salinity in the Delta.

## 2. Materials and Methods

### 2.1. Study Area

The Yellow River Delta is located in the northeast part of China's Shandong Province. It is composed of sediments carried by China's Yellow River and deposited at the river's mouth, creating the estuary. The Delta is surrounded by the Bohai Sea to the east and north as shown in Figure 1. It has the highest speed of land growth in the world, exceeding on average 13.8 km$^2$ per year, which make it the main production area of grain and an important reserve land resource in China [29]. The earliest development of YRD can be traced back to the late 1960s when the petroleum resource was firstly discovered in this region. The second largest oil field (Shengli oil field) is located here. Yellow River Delta ranks in a high position in the global ecosystem, for it provides an important staging, wintering and breeding site for birds in Northeast Asia and around the Pacific migration route, and it is also a breeding site for some rare birds which are endangered in the world. The nature reserve of YRD was founded by the State Council in 1992. As the region's coastal wetland-aquifer systems are fragile and vulnerable to environmental changes (e.g., seawater intrusion, freshwater shortage, salinization and land subsidence), it is important to understand the hydrological processes impacting the groundwater system of the YRD.

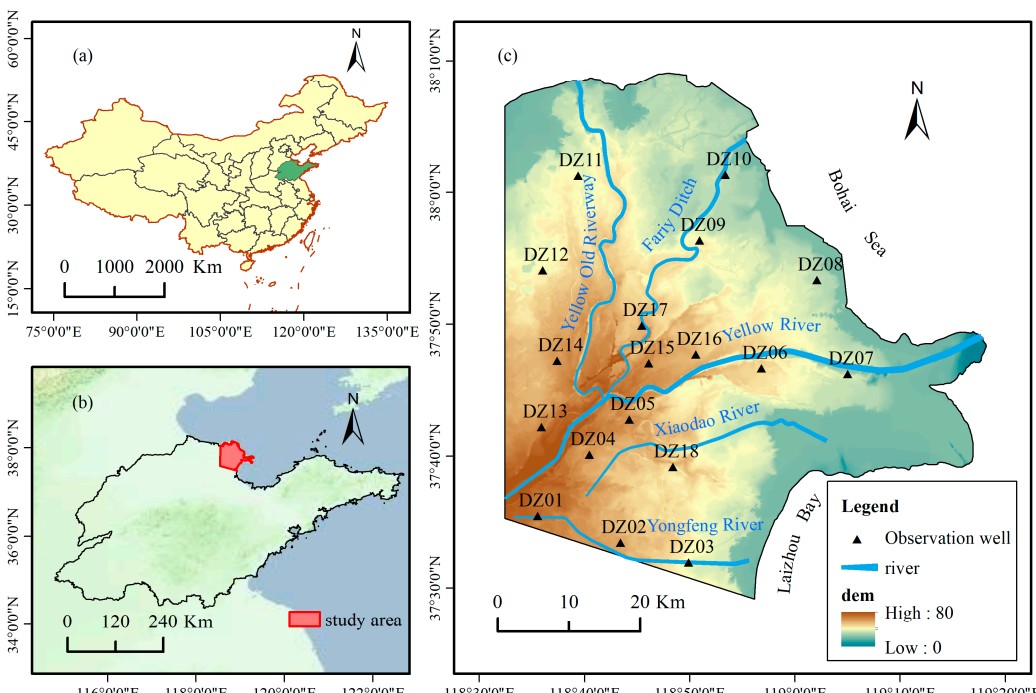

**Figure 1.** Study area description: (**a**) Shandong Province in the map of China; (**b**) location of the Yellow River Delta in the map of Shandong Province; (**c**) the distribution of groundwater observation wells in the Yellow River Delta.

Three distinct delta zones can be discerned in the history of Yellow River Delta development. At the basis is the ancient Yellow River Delta which was formed before 1855 with an area of 6000 km$^2$. Further downstream, the new Yellow River Delta formed from 1855 with an area of 5400 km$^2$. The downmost zone is the contemporary Yellow River Delta with an area of 2800 km$^2$, which has developed since 1934. The study area of this paper is concerned with the region of the modern Yellow River Delta (Figure 1).

The modern Yellow River Delta can be defined as a lowland coastline area with the altitude ranging from 0 to 10 m above sea level. The surface topography is in low-gradient, elevation decreases from south-west to north-east with ratios varies from 1/8000 to 1/12,000. The elevation of the YRD also reflects the sedimentation history of shifting river channels. Each alluvial fan is highest in its central and upstream with 3–5 m higher

than around lowland. This area has a monsoon climate of the warm-temperate zone with a characteristic of abundant sunlight and obvious seasonal changes of temperature, humidity and evaporation. The average annual precipitation is 530–630 mm, most of which is rainfall during summer. Average evaporation is 1900–2400 mm. The evaporation value is three times larger than that of the rainfall, which is the natural prerequisite for the occurrence of soil and groundwater salinity.

Shallow groundwater in the YRD is mainly stored in Quaternary loose sediments, including fluvial faces, flood plain faces and marine sediments, with a maximum depth of approximately 70 m. The sediment thickness becomes thinner from inland to the coastline (the river mouth). The lithology of sediments is mainly silt, fine sand, clay and sub-clay. The YRD is largely defined by a plateaued, flat terrain (see Figure 1) and the shallow groundwater hydraulic gradient averages 1/7000. Accordingly, the horizontal component of groundwater flow in the region is very slow, while the vertical exchange is dramatic [29]. For example, the average annual rainfall on the Delta is 530–630 mm while the mean annual potential evaporation in the area is 1900–2400 mm [30]. As commonly associated with marine environments, saline groundwater (TDS 5–20 g/L) is widely distributed within the Delta. Some fresh groundwater lens store beneath the Yellow River channels [31].

### 2.2. Groundwater Observation Wells

In an attempt to monitor groundwater dynamics, 18 boreholes (to a depth of 6 m) were located throughout the delta with an average spacing of about 8 km to monitor shallow groundwater fluctuations and electrical conductivity variance. Most of the boreholes were distributed along lines extending from the centre of the Delta to the coast. The locations of the boreholes are shown in Figure 1. The borehole casings consist of 50 mm diameter PVC pipe with a 2 m length of continuous slotted screen at the bottom. Groundwater monitoring continued from May 2004 to June 2010. Groundwater depth and EC values have been measured and recorded at 5-day intervals for a period of 6 years, except for monitoring well DZ07. Because the location of this well is very close to the Yellow River channel, it was flooded and abandoned after May 2007.

In our previous research, the total dissolved solids TDS (g/L) and the concentration of chloride ions Cl (g/L) of groundwater samples were highly related to their ECs in YRD; see Equations (1) and (2). Here, we use the groundwater EC value (μs/cm) to represent the salinity of groundwater because it is reliable and convenient for analysis. Groundwater EC was measured using a H-BD5W handheld conductivity meter.

$$EC = 52.95 \ln TDS + 866.45 \left( R^2 = 0.98 \right) \tag{1}$$

$$EC = 49.89 \ln Cl + 910.31 \left( R^2 = 0.98 \right) \tag{2}$$

In recent years, various methods have been used to estimate and evaluate seawater intrusion and its impact on groundwater, and many studies have used total dissolved solids (TDS) and Cl ion to determine the extent of seawater intrusion [32,33]. According to the "technical specification for seawater intrusion monitoring and assessment" published by the State Oceanic Administration of China in 2014, groundwater TDS exceeding 1 g/L is considered mild seawater intrusion, and 2 g/L is considered severe seawater intrusion. These two values are equivalent to the EC values of 866 and 903 μs/cm, and we use these standards to judge the severity of seawater intrusion in this study.

### 2.3. Other Data

Other ancillary data include: digital elevation model (DEM), rainfall and evaporation records, etc. The DEM data with 10 m resolution were digitised and gridded from 1: 10,000 scale contour line maps. The daily rainfall and evaporation records from 2004 to 2010 were collected from 4 local weather stations with or near the study area.

*2.4. Research Methodology*

2.4.1. Exploratory Analyses and Spatial Interpolation

The relative groundwater level was calculated using the measured groundwater level minus 1.17 m (the annual mean sea level observed by the local hydrological station). Here, we use the relative groundwater level of each site for interpolation because it can obviously and directly show where the groundwater level is below or above the local sea level and indicate the extent of potential seawater intrusion to identify the seasonal variations in groundwater levels and the distribution pattern of EC. Here, we design three scenarios for analysis, in which the mean, 5th and 95th percentiles of groundwater level and EC at each monitoring site were calculated and used for interpolation. Considering the data outliers, the 5th and 95th percentiles are used in this study instead of the maximum and minimum values. The Kriging method was used to interpolate groundwater level and EC from site to regional scale and map their distribution patterns.

2.4.2. STL Decomposition Method

Repetitive measurements of groundwater levels track persistent and transient changes in the water levels through time. Persistent changes can include trend and seasonal components that characterise the groundwater resource and provide information needed for assessment and effective management. Ephemeral changes reflect daily changes that are influenced by recent weather conditions, such as the time since the last soaking rain [34]. STL uses locally weighted regression (LOESS) to decompose the seasonal component [35]. To estimate the trend component of a non-seasonal time series that can be described by an additive model, it is common to use a smoothing method, such as calculating the simple moving average of the time series. In order to obtain the features of groundwater levels and TDS dynamics, the seasonal-trend decomposition method based on Loess (STL) was employed to each time-series records of groundwater level and salinity at all sites. The general model of this method [36] can be described as:

$$Y_t = T_t + S_t + R_t \tag{3}$$

where $Y_t$ is the groundwater level or sanity at time $t$, $T_t$ is the trend component; $S_t$ is the seasonal component; and $R_t$ is an irregular (residual) component.

It is necessary to set the smoothing parameters for the trend and seasonal components when using the STL decomposition. Different choices of smoothing parameters were experimented for all the observation wells, and we finally choose window widths of 1 year in both trend and seasonal components for all subsequent STL analyses.

2.4.3. Trending Analysis and Correlation Analysis

The Mann-Kendall's trend test and Sen's slope estimates for trends are widely used for analyzing groundwater monitoring data [37,38]. In this research, the trending components of GWL and EC decomposed by STL were used for MK analysis instead of original time-series records, to avoid the impact of periodic seasonal changes to the testing results.

The Mann-Kendall statistic ($S$) is calculated as shown in Equation (4):

$$S = \sum_{i=1}^{n-1} \sum_{j=i+1}^{n} sgn(x_j - x_i), \ where \ sgn(x_j - x_i) = \begin{cases} 1, \ x_j - x_i > 0 \\ 0, \ x_j - x_i = 0 \\ -1, \ x_j - x_i < 0 \end{cases} \tag{4}$$

where $x_i$ and $x_j$ are the sequential data values of measured groundwater levels and n is the record length. The variance is computed as:

$$Var(S) = \frac{n(n-1)(2n+5) - \sum_{i=1}^{m} t_i(t_i - 1)(2t_i + 5)}{18} \tag{5}$$

where $m$ is the number of tied groups and $t_i$ denotes the number of ties of extent $i$. The standard normal test statistic $Z_s$ is calculated by Equation (6):

$$Z_s = \begin{cases} \frac{S-1}{\sqrt{Var(S)}}, & if \ S > 0 \\ 0, & if \ S = 0 \\ \frac{S+1}{\sqrt{Var(S)}}, & if \ S < 0 \end{cases} \tag{6}$$

Positive values of $Z_s$ indicate increasing trends, while negative $Z_s$ values indicate decreasing trends. For the purpose of this study, a significance level $\alpha = 0.05$ is used. The null hypothesis of no trend is rejected if $|Z_s| > 1.96$.

The slope of groundwater level measurements is estimated as described by Sen. The Sen's slope estimator ($Q_i$) is calculated as shown by Equation (7).

$$\left( Q_i = \frac{x_j - x_k}{j - k} \right) for \ i = 1, 2, \cdots, n \tag{7}$$

Sen's trend estimator is the median of the estimates of the slope. The positive value of Qi indicates an increasing trend of time series, while a negative value indicates a decreasing trend.

The relationship between the seasonal components of groundwater and influencing factors such as rainfall and evapotranspiration was determined using a Pearson correlation analysis.

## 3. Results

### 3.1. The Spatial Distributions of Groundwater Table and EC in Different Scenarios

The interpolated groundwater level maps are shown in Figure 2. The basic distribution pattern of the groundwater level is that the groundwater level gradually decreases from the southwest inland to the north and east coast. The negative values in Figure 2a–c show the area where the local groundwater level is below the local sea level, and the positive values represent where the groundwater level is higher than the local sea level. For the scenario of the mean groundwater level (Figure 2a), the groundwater level is lower than the local sea level in up to 42% of the region. These areas are mainly along the coastline. Groundwater levels show clear seasonal variations. Compared to Figure 2a, the lowest (5% percentiles) groundwater level (Figure 2b) generally decreases by about 0.859 m, and it shows that more than 66% of the lowest groundwater levels are lower than the local sea level in the dry season. The highest (95% percentiles) groundwater level is about 0.982 m higher than the mean groundwater level, and about 19% of the groundwater levels are lower than the local sea level.

The spatial distribution of groundwater EC is shown in Figure 2d–f. The basic distribution pattern of the groundwater EC is that the groundwater gradually becomes more saline from the inland area close to the river to the coastal area. However, in the western and central parts of the study area, there are some areas of highly saline brine groundwater that are sparsely distributed. Most of these areas are in the depressions between the river channels. The relatively fresh groundwater is mainly buried under the present and old Yellow River channels. However, the freshwater lens is unstable and easily degraded, and the extent of fresh groundwater has shrunk dramatically during the dry seasons. The area with different severity of seawater intrusion was calculated and shown in Table 1. It seems that the seawater intrusion shallow groundwater in this region. For the average groundwater EC scenario, about 96% of the region has moderate or severe seawater intrusion. The extent and severity of seawater intrusion is highly dynamic. The most saline groundwater is distributed in the northeast of the coastline, near observation wells dz08 and dz10, the groundwater EC can be higher than the local seawater EC (TDS = 31.45 g/L, EC = 1046 μs/cm) in Figure 2c.

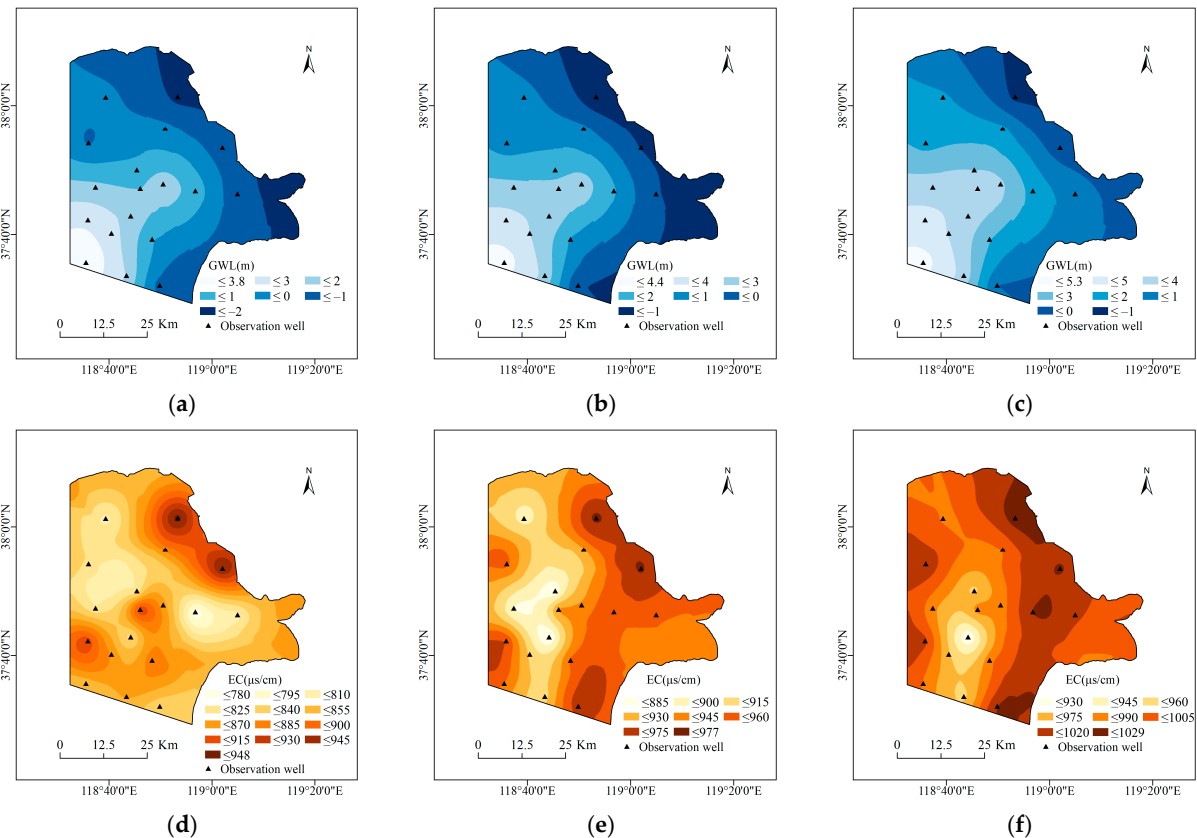

**Figure 2.** The distributional maps of groundwater level (GWL) and electrical conductivity (EC) in three scenarios: (**a**) 5% percentiles groundwater levels; (**b**) mean groundwater levels; (**c**) 95% percentiles groundwater levels; (**d**) 5% percentiles groundwater ECs; (**e**) mean groundwater ECs; (**f**) 95% percentiles groundwater ECs.

**Table 1.** The area with different severities of seawater intrusion according to groundwater EC.

|  | No Seawater Intrusion | Mild Seawater Intrusion | Severe Seawater Intrusion |
|---|---|---|---|
| 5% percentiles | 79% | 14% | 7% |
| mean value | 4% | 15% | 81% |
| 95% percentiles | – | – | 100% |

### 3.2. The Trending Characteristics of Groundwater Level and EC at Each Sites

The trending components of groundwater levels and ECs decomposed by STL at each site are shown in Figures A1 and A2. The results of the MK analysis for the trending components are shown in Table 2. It shows that most of the observation wells (16 out of 18) have the decreasing trend of groundwater level, with the decreasing rate of 0.045 m/a~0.443 m/a. Only the observation wells (DZ09 and DZ11) near the reservoir and the surrounding mariculture ponds in the northern part of the Delta have the opposite trend of increasing local groundwater level at the rate of 0.09 m/a~0.21 m/a. Half of the observation wells (9 out of 18) have the increasing trends in groundwater EC, with the increasing rate of 1.153~25.608 μs/cm.a. Most of them are located near the coast or the new estuary of the Yellow River. Moreover, there are some inland sites where groundwater EC have the significant saline trends, e.g., DZ12 and DZ13. Only two observation wells have no obvious trend, including DZ09 and DZ15. The other 7 observation wells have the decreasing trends in groundwater EC, the decreasing rate ranges between 1.829~12.677μs/cm.a.

Most of these decreasing wells are located at relatively higher elevation and their land use includes farmland, residential area or forest, where the higher frequency of human activities may induce better irrigation and drainage condition that can improve the local groundwater environment.

**Table 2.** The results of MK analysis for trending components of groundwater level and EC.

| Title 1 | Groundwater Level | | | Groundwater EC | | |
|---|---|---|---|---|---|---|
| | MK_z Value | Trend | Sen's Slope | MK_z Value | Trend | Sen's Slope |
| DZ01 | −21.800 | decreasing | −0.050 | −9.957 | decreasing | −2.813 |
| DZ02 | −20.868 | decreasing | −0.105 | −14.171 | decreasing | −3.589 |
| DZ03 | −3.925 | decreasing | −0.014 | 6.365 | increasing | 2.864 |
| DZ04 | −13.018 | decreasing | −0.081 | −0.022 | no trend | −0.086 |
| DZ05 | −3.837 | decreasing | −0.050 | 11.953 | increasing | 2.411 |
| DZ06 | −12.796 | decreasing | −0.061 | 19.804 | increasing | 14.278 |
| DZ07 | −13.620 | decreasing | −0.259 | 3.566 | increasing | 7.094 |
| DZ08 | −20.868 | decreasing | −0.058 | 8.804 | increasing | 2.660 |
| DZ09 | 20.957 | increasing | 0.208 | 1.663 | no trend | 1.177 |
| DZ10 | −15.989 | decreasing | −0.161 | 6.010 | increasing | 1.530 |
| DZ11 | 10.933 | increasing | 0.099 | −15.191 | decreasing | −11.600 |
| DZ12 | −18.429 | decreasing | −0.315 | 12.885 | increasing | 25.608 |
| DZ13 | −13.506 | decreasing | −0.096 | 16.034 | increasing | 10.756 |
| DZ14 | −10.356 | decreasing | −0.121 | −9.469 | decreasing | −4.585 |
| DZ15 | −19.582 | decreasing | −0.443 | 0.067 | no trend | 0.019 |
| DZ16 | −11.598 | decreasing | −0.077 | 20.203 | increasing | 13.520 |
| DZ17 | −8.253 | decreasing | −0.045 | −18.562 | decreasing | −12.677 |
| DZ18 | −12.397 | decreasing | −0.083 | −7.074 | decreasing | −1.829 |

The spatial distribution of groundwater level and EC development trends are shown in Figure 3. Combining the trends of groundwater level and EC, Figure 3 shows that the groundwater along the coast was relatively stable, with moderate trends of groundwater level decrease and EC increase. The groundwater south of the Yellow River has the tendency of groundwater improvement, because these areas have the relatively long history of human activities and cultivation. The inland area, which suffers from groundwater degradation with decreasing groundwater level and increasing groundwater EC (Figure 3), have the relatively shallow groundwater depth and some of them have been surrounded by oil production wells (DZ06, DZ16, DZ12). The oil production activities may cause the deep brine to contaminate the shallow groundwater and increase the groundwater EC.

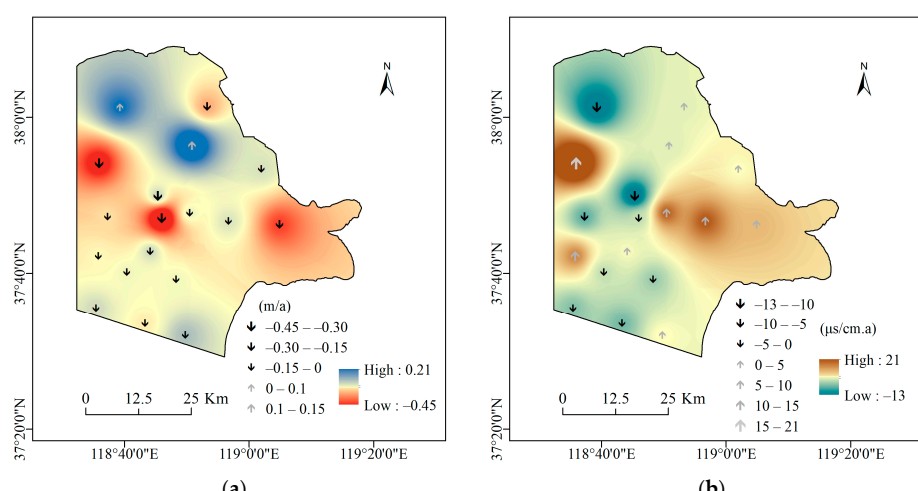

**Figure 3.** The distributional maps of the developing trends of groundwater: (**a**) GWL; (**b**)EC.

*3.3. The Seasonal Characteristics of Groundwater Level and EC*

The seasonal components of groundwater level and EC decomposed by STL at each site are shown in Figures A3 and A4. All monitoring wells have similar seasonal characteristics in groundwater level variation. In the winter months of January and February, the groundwater depth reaches its maximum, and the groundwater slowly rebounds in the spring. With the arrival of the rainy season, the groundwater level peaks in July and August and then gradually declines. However, some observation wells have more obvious seasonal variations, while other observation wells have weaker seasonal variations (such as DZ08, DZ10, DZ06 and DZ16). These monitoring sites are mainly located in coastal areas or along the Yellow River, and the seasonal effects of precipitation and evaporation on groundwater are weakened by the influence of river and ocean factors.

Most of the wells have obvious seasonal variance in groundwater EC. However, when compared with groundwater levels, their seasonal variation in groundwater EC was more complicated. About half of the wells show a significant increase in groundwater EC during the summer season (July to September), e.g., DZ01, DZ02, DZ05, DZ08-DZ10, DZ12 and DZ15–17, because their seasonal components are sensitive to evaporation (Table 3). Some wells, e.g., DZ03, DZ06, DZ07 and DZ14, have the opposite dynamic feature in groundwater EC, with their groundwater EC decreasing in the summer seasons. It seems that the freshwater recharge by rainfall in the summer season can temporarily desalinate the groundwater. The other wells (DZ09, DZ11, DZ13) have the weak seasonal variation in groundwater EC, which is not sensitive to rainfall or evaporation.

**Table 3.** The correlation analysis results between the seasonal components of groundwater level and EC with the seasonal components of rainfall and evaporation.

| | Groundwater Level | | Groundwater EC | |
|---|---|---|---|---|
| | **Rainfall** | **Evaporation** | **Rainfall** | **Evaporation** |
| DZ01 | 0.58 ** | 0.632 ** | −0.063 ** | 0.437 ** |
| DZ02 | 0.452 ** | 0.362 ** | 0.484 ** | 0.698 ** |
| DZ03 | 0.512 ** | 0.502 ** | −0.391 ** | −0.296 ** |
| DZ04 | 0.637 ** | 0.509 ** | 0.416 ** | 0.342 ** |
| DZ05 | 0.301 ** | 0.602 ** | 0.404 ** | 0.789 ** |
| DZ06 | 0.532 ** | 0.275 ** | −0.597 ** | −0.176 ** |
| DZ07 | 0.646 ** | 0.278 ** | −0.586 ** | −0.264 ** |
| DZ08 | 0.458 ** | 0.238 ** | 0.396 ** | 0.7 ** |
| DZ09 | 0.453 ** | 0.05 | 0.5 ** | 0.503 ** |
| DZ10 | 0.49 ** | 0.131 ** | 0.397 ** | 0.674 ** |
| DZ11 | 0.55 ** | 0.08 | −0.272 ** | −0.506 ** |
| DZ12 | 0.249 ** | 0.047 | 0.323 ** | 0.569 ** |
| DZ13 | 0.521 ** | 0.294 ** | 0.333 ** | 0.411 ** |
| DZ14 | 0.583 ** | 0.555 ** | −0.323 ** | 0.19 ** |
| DZ15 | 0.519 ** | 0.037 | 0.335 ** | 0.656 ** |
| DZ16 | 0.31 ** | −0.222 ** | 0.425 ** | 0.775 ** |
| DZ17 | 0.196 ** | −0.143 ** | 0.176 ** | 0.557 ** |
| DZ18 | 0.582 ** | 0.559 ** | 0.205 ** | 0.253 ** |

Note: ** Statistically significant at 1% level of significance.

The results of the correlation analysis between the seasonal components of groundwater level and groundwater EC with the seasonal components of rainfall and evaporation are shown in Table 3. It shows that the seasonal components of groundwater levels are significantly correlated with rainfall, which can recharge the shallow groundwater and generally raise the groundwater level in summer. However, the groundwater levels of some wells are positively correlated with evaporation because the rainfall and high evaporation occurred in the same season. Furthermore, in the shallow aquifer, water recharge from rainfall exceeds water use from evaporation, causing groundwater to rise during the rainy season. The seasonal components of groundwater EC have a positive relationship with the

seasonal components of evaporation in most of the wells, i.e., evaporation can condense the shallow groundwater and increase its salinity. The groundwater EC in some wells have a negative relationship with rainfall, as it seems that the groundwater EC in these sites are sensitive to rainfall and the groundwater salinity can be temporarily improved during the rainy season. The groundwater EC in the other wells are not sensitive to rainfall.

## 4. Discussion

### 4.1. The Threaten of Seawater Intrusion in YRD

Most studies of aquifers affected by seawater intrusion have focused on either groundwater level or groundwater salinity [39–41]. Ismail Abd-Elaty et al. combined the pumping from the delta aquifer and meteorological data and found that groundwater level decline in the aquifer and sea level rise would exacerbate seawater intrusion [42]. We have combined the analysis of groundwater level and salinity to accurately reflect the extent and magnitude of seawater intrusion. Our results show that the threat of seawater intrusion of YRD is rising during 2004–2010, and the regionally averaged groundwater level is lower than the local sea level, especially in the coastal area where the groundwater level is lower than the sea level, with the difference value of 1–2 m. Combining the distribution maps of groundwater level and EC, it can be seen that the area most prone to seawater intrusion is located in the northeastern part of the coastline, not only because of its low elevation, but also because the largest oil field (Gudong oil field) of the region is located there, and the oil field activity may change the groundwater flow direction and increase the risk of seawater intrusion.

There is a close relationship between climate change and water resources [43]. Hydrometeorological factors have certain influence on the seasonal variation of groundwater. Chaoxia Lu et al. showed that precipitation reduces groundwater depth in the summer [44]. This study combined precipitation and evaporation data to analyze the dynamic change of groundwater in the YRD. Our results showed that the shallow groundwater of the YRD is sensitive to changes in rainfall and evaporation. Drought may lead to degradation of the groundwater environment and increase the threat of seawater intrusion. The YRD has a high evaporation-precipitation ratio, a heavy degree of salinization, and it also has a significantly increasing trend in the occurrence frequency and degree of seasonal drought [45]. In addition, from 1980 to 2021, the rate of sea level rise in the Bohai Sea was 3.6 mm per year, higher than the global average for the same period [46]. It is estimated that the sea level along China's coast will rise by 68 to 170 mm in the next 30 years. Both drought and sea level rise may accelerate seawater intrusion and salinization of groundwater and soil, so it is very important to continue to monitor and analyze groundwater dynamics.

### 4.2. Human Activities' Impact on Groundwater Development

Human activities in the YRD have been increasing over the past twenty years. The Yellow River provides a reliable and steady freshwater resource to this region, making it the most important reserved arable land resource in China. While the widely distributed wetlands and mudflats can be easily developed into aquaculture ponds, human activities such as the construction of reservoirs and aquaculture ponds can alter the natural distribution of water resources and increase surface water recharge to shallow groundwater [42,47]. Irrigation and drainage activities can improve the salinity of the soil and shallow groundwater. Our analyses indicate that groundwater around agricultural and residential areas is less saline than groundwater around wasteland and grassland.

Oil production has a long history in the YRD, as it is home to the second-largest oil field in China [48]. Apart from the largest Gudong oil field, which is concentrated to the northeast of the coastline, there are numerous oil wells scattered throughout the interior [49]. It is reported that oil exploitation in the YRD has increased dramatically and some of the abandoned oil wells have been reused for brine extraction. Deep groundwater extraction can affect the natural exchange between shallow and deep groundwater, and even cause seawater intrusion [50]. Our research shows that groundwater around oil wells

tends to deteriorate. However, the impact of oil production on groundwater needs to be further investigated through field surveys and monitoring.

## 5. Conclusions

Based on groundwater level and EC records from 2004 to 2010, this research analyzed the spatio-temporal dynamic characteristics of shallow groundwater and its influencing factors in YRD. Interpolated maps of groundwater levels and ECs in different scenarios were generated to characterize the spatial patterns of groundwater levels and ECs, and to reveal the degree of seawater intrusion in this region. It shows that the mean groundwater level over 42% of the region was lower than the local sea level, while the mean groundwater EC over 96% of the region met the standards for seawater intrusion. Although the fresh groundwater lens existed under the Yellow River channels, it was easily and rapidly salinized during the dry seasons.

The time series of groundwater level and EC at each site were decomposed using a robust seasonal trend decomposition (STL) technique. The results of STL analysis for the trend components of groundwater level and EC indicate that groundwater level generally declined by 0.01~0.45 m/a, and local groundwater level increases of 0.09 m/a~0.21 m/a were detected near mariculture ponds and reservoirs in the northern part of the Delta. The trends of groundwater EC are more complex: about half of the wells have the increasing trend in groundwater EC, while some wells located in the central of the region have the rapid saline speed. The oil exploration activities may cause the shallow groundwater to be salinized by deep brine.

All studied stations, however, showed that seasonal variability of the groundwater was strongly correlated with rainfall and evaporation. The Pearson correlation coefficient values between seasonal components of groundwater level and precipitation vary from 0.2 to 0.65. The average correlation coefficient values between groundwater EC and evaporation vary from −0.5 to 0.79. The intensive rainfall in summer can raise the water table and temporarily desalinate the groundwater. The seasonal components of groundwater ECs were sensitive to evaporation, so the prolonged drought may degrade the groundwater resource and related ecosystem; thus, the timely supply of fresh water during the dry season is important.

Under the situation of sea level rise and lack of freshwater resources, groundwater monitoring should be continued and intensified in an attempt to protect the groundwater resource and the sustainability of local agriculture and the ecosystem. For the management of natural resources, it is important to ensure the inflow of the Yellow River estuary and optimize the spatial allocation of water resources in this region. The impact of human activities on groundwater dynamics is complex; therefore, further survey and analysis will be carried out in our future works.

**Author Contributions:** Methodology, X.F.; software, T.M.; writing—original draft preparation, X.F. and T.M.; writing—review and editing, X.F., T.M. and X.D.; project administration, X.F.; funding acquisition, X.F. All authors have read and agreed to the published version of the manuscript.

**Funding:** This study was sponsored by the National Natural Science Foundation of China (No. 41971107).

**Data Availability Statement:** The data that support the findings of this study are available from the first author upon reasonable request.

**Acknowledgments:** The authors are very grateful to the Institute of Geographic Sciences and Resources, Chinese Academy of Sciences for providing observation data on groundwater, and also to the financial support of the National Natural Science Foundation of China.

**Conflicts of Interest:** The authors declare no conflict of interest.

## Appendix A

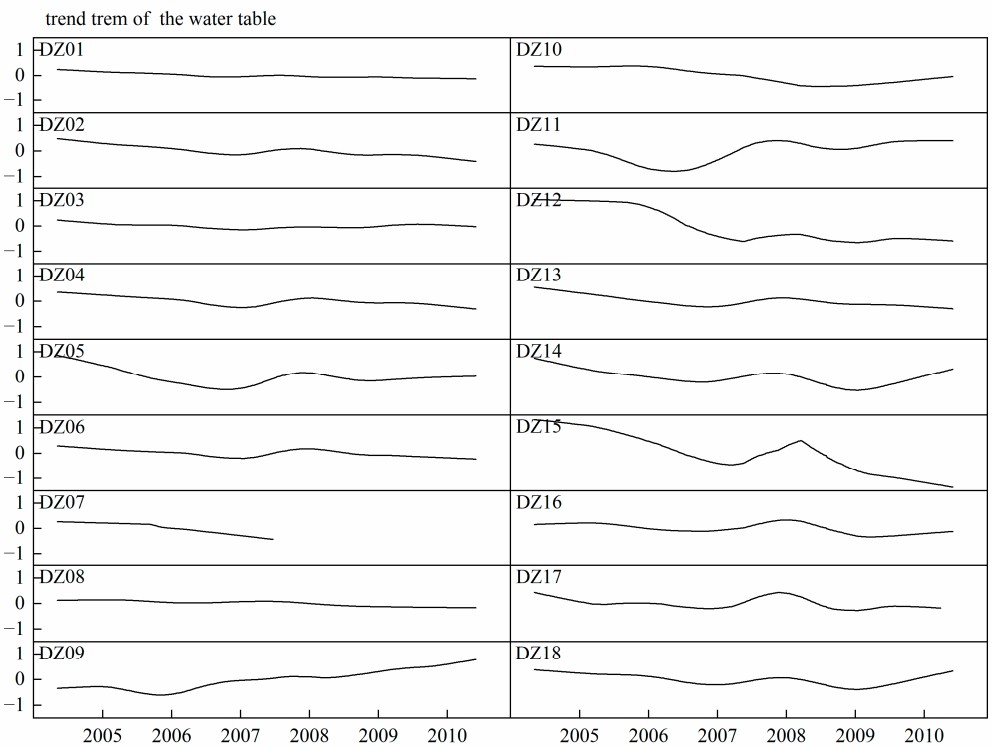

**Figure A1.** The trend term of GWL of each observational well varies with time.

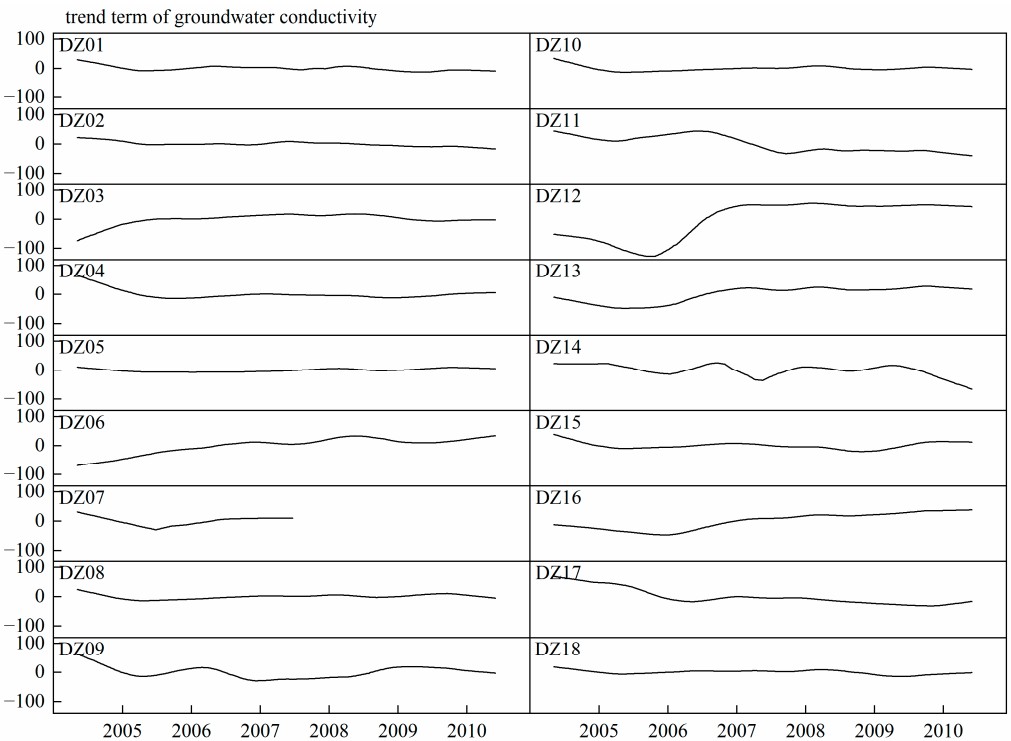

**Figure A2.** The trend term of groundwater EC of each observational well varies with time.

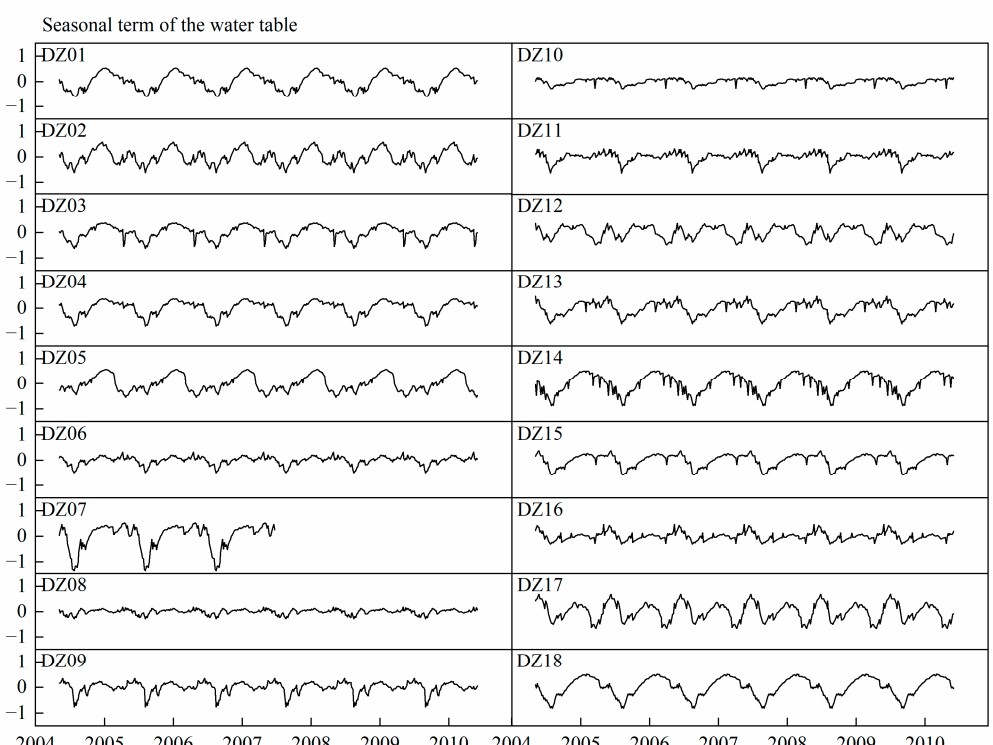

**Figure A3.** The seasonal term of GWL of each observational well varies with time.

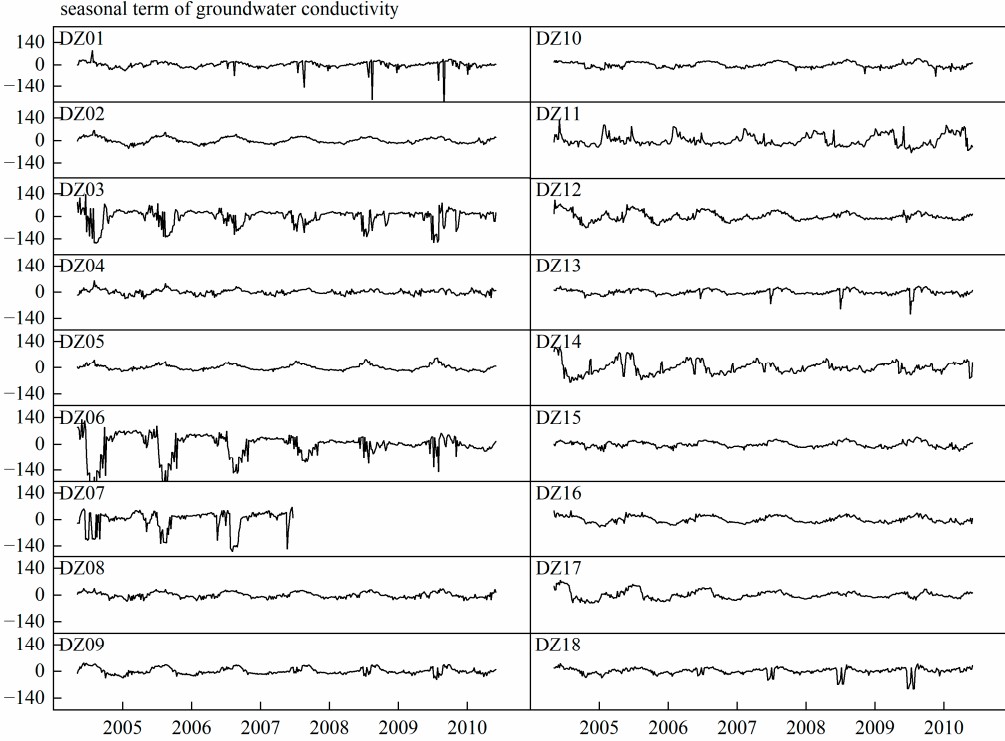

**Figure A4.** The seasonal term of groundwater EC of each observational well varies with time.

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
