# Peer review of "The Spatio-Temporal Dynamic Patterns of Shallow Groundwater Level and Salinity: The Yellow River Delta, China"

_water, doi:10.3390/w15071426_

Round 1

Reviewer 1 Report

General Comments

The authors designed the article to study the “The spatio-temporal dynamic patterns of

shallow groundwater level and salinity: The Yellow River Delta, China.

The authors tried very well to explain their designed research. The article discrepancies are

mentioned to increase the quality of the article before final publication.

Specific Comments

The article introduction should be revised because the designed methodology, data collection and the key results of the article are not mentioned in the abstract. Kindly revise it.

Please revise the introduction section and add the literature about seasonal variability of the costal water. The Iqbal et al., published many article on the seasonal water quality. Kindly take guidelines from their article and also cite them properly.

Iqbal, Muhammad Mazhar, et al. "Analysis of Seasonal Variations in Surface Water Quality over Wet and Dry Regions." Water 14.7 (2022): 1058.

Iqbal, Muhammad Mazhar, et al. "Seasonal effect of agricultural pollutants on coastline environment: a case study of the southern estuarine water ecosystem of the boseong county Korea." Pakistan Journal of Agricultural Sciences 59.1 (2022).

Rizvi, Filza Fatima, et al. "Assessment of climate extremes from historical data (1960-2013) of Soan River Basin in Pakistan." International Journal of Global Warming 25.1 (2021): 1-37.

Kindly adjust the upper case and lower case mistakes in your article, see L88, L84, L102-104, etc. Check it throughout the article.

Revise the caption of figure-1. Mention the detailed explanation and also mention a, b, and c on the figures and also explain them accordingly.

The discussion of the article is very week and not properly cited with the literatures. Kindly add more citations in the discussion section.

The conclusion section needs to be revised with the quantification strategy.

Reviewer 2 Report

Key comments

1.      In the keywords what is the abbreviation of GWL, EC, STL.

2.      What are the limitations of the measurements and methods used to quantify the relationship between aquifer flow and the seawater interface?

3.      What are the key influences on groundwater flow and seawater intrusion?

4.      Can geophysics reveal the saline groundwater wedge, and can it assist in the validation of groundwater modeling? 

Round 2

Reviewer 1 Report

accepted.